# Fas/CD95 Signaling Pathway in Damage-Associated Molecular Pattern (DAMP)-Sensing Receptors

**DOI:** 10.3390/cells11091438

**Published:** 2022-04-24

**Authors:** Gael Galli, Pierre Vacher, Bernhard Ryffel, Patrick Blanco, Patrick Legembre

**Affiliations:** 1CNRS, ImmunoConcEpT, UMR 5164, University Bordeaux, 33000 Bordeaux, France; ggalli@immuconcept.org (G.G.); patrick.blanco@chu-bordeaux.fr (P.B.); 2Centre National de Référence Maladie Auto-Immune et Systémique Rares Est/Sud-Ouest (RESO), Bordeaux University Hospital, 33076 Bordeaux, France; 3Department of Internal Medicine, Haut-Leveque, Bordeaux University Hospital, 33604 Pessac, France; 4INSERM, CRCTB, U1045, University Bordeaux, 33000 Bordeaux, France; pierre.vacher@inserm.fr; 5CNRS, INEM, UMR7355, University of Orleans, 45071 Orleans, France; bernhard.ryffel@cnrs-orleans.fr; 6UMR CNRS 7276, INSERM U1262, CRIBL, Université Limoges, 87025 Limoges, France

**Keywords:** apoptosis, auto-immunity, CD95, inflammasome, necroptosis, pyroptosis, sting

## Abstract

Study of the initial steps of the CD95-mediated signaling pathways is a field of intense research and a long list of actors has been described in the literature. Nonetheless, the dynamism of protein-protein interactions (PPIs) occurring in the presence or absence of its natural ligand, CD95L, and the cellular distribution where these PPIs take place render it difficult to predict what will be the cellular outcome associated with the receptor engagement. Accordingly, CD95 stimulation can trigger apoptosis, necroptosis, pyroptosis, or pro-inflammatory signaling pathways such as nuclear factor kappa-light-chain-enhancer of activated B cells (NF-κB) and phosphatidylinositol-3-kinase (PI3K). Recent data suggest that CD95 can also activate pattern recognition receptors (PRRs) known to sense damage-associated molecular patterns (DAMPs) such as DNA debris and dead cells. This activation might contribute to the pro-inflammatory role of CD95 and favor cancer development or severity of chronic inflammatory and auto-immune disorders. Herein, we discuss some of the molecular links that might connect the CD95 signaling to DAMP sensors.

## 1. Introduction

Although CD95 (Fas) has long been viewed as the death receptor prototype, only involved in the induction of the apoptotic signaling pathway, more recent and accumulating evidence point out that this transmembrane receptor contributes to chronic inflammatory disorders and cancer by inducing non-apoptotic signaling pathways [1,2]. Herein, we first discuss the pathologies associated with genetic mutations in the CD95 signaling pathway and in damage-associated molecular patterns (DAMP) sensors. Second, we expose the different links existing between CD95 or its downstream apoptotic machinery including the adaptor protein Fas-Associated protein with Death Domain (FADD) and the initiator protease caspase-8 in the activation/regulation of the DAMP-sensing complexes. The main goal of this review is to question whether CD95 might act as an alarmin for Pattern Recognition Receptors (PRRs).

## 2. CD95 and DAMP-Sensor-Mediated Pathologies

### 2.1. Genetic Mutations in CD95 and DAMP Sensors and Chronic Inflammatory Diseases

#### 2.1.1. CD95-Associated Genetic Disorders

In humans, CD95 mutation leads to the development of a disease called auto-immune lymphoproliferative syndrome or ALPS, also known as Canale–Smith syndrome [3]. ALPS involves the development of early onset polyclonal lymphoproliferation (splenomegaly, adenopathy) associated with expansion of a population of aberrant double-negative B220^+^CD3^+^TCRαβ^+^CD4^−^CD8^−^ T cells. To qualify as ALPS, the evolution of the disease must be superior to 6 months and with exclusion of a secondary etiology for the lymphoproliferation (hematological malignancies or solid cancers, for example) [4,5]. This cardinal presentation is associated with mutation of CD95, CD95L, or caspase 10, a defect in lymphocyte proliferation, autoimmune cytopenia (mainly autoimmune hemolytic anemia and immune thrombocytopenic purpura), augmentation of Immunoglobulin A and G titers, associated with elevated dosage of B12 vitamin, augmentation of IL-10 or IL-18 and an increase in soluble-CD95L in patient serum. Of note, ALPS present an increased risk of hematological malignancies, which can complicate its diagnosis [6]. ALPS are divided into five subtypes, showing mutations in CD95, CD95L, or caspase-10 (see Table 1). Although most of the ALPS patients exhibit CD95 mutations, when Oliveira et al. revised the ALPS classification in 2010, lymphoproliferative disorders showing no mutations in CD95, CD95L, or caspase 10 were also classified as ALPS disorders (i.e., caspase 8, NRAS/KRAS, or SH2D1a mutations). In addition, Dianzani Autoimmune Lymphoproliferative Disease (DALD) was also reported as an ALPS-related disorder even though still no genetic defect was identified [4,5].

Similarly, CD95 (Lpr and Lpr^cg^ mice) or CD95L (Gld) mutations in mice lead to a similar phenotype with aberrant T cell proliferation and lymphoproliferation, but also arthritis and early death from acute glomerulonephritis. These mice exhibit the development of autoantibodies, such as those found in Systemic Lupus Erythematosus (SLE) patients, and have led to their use as a model for SLE physio-pathological studies. However, this autoantibody development in CD95-mutated mice represents a striking difference with the human pathology in which autoantibodies do not represent a common clinical feature of human ALPS [7].

#### 2.1.2. DAMP-Sensor-Associated Genetic Disorders

DAMP are non-infectious triggers of the immune system released by dying or damaged cells [8] and detected by various membrane-bound or intracellular sensors, notably through a cytosolic complex called inflammasome [9]. These DAMP sensors have been implicated in several inflammatory diseases, on a spectrum ranging from autoimmune polygenic phenotypes such as in systemic lupus erythematosus, rheumatoid arthritis or Crohn’s disease, to monogenic autoinflammatory disorders [10].

In these monogenic autoinflammatory syndromes, enhanced activation of the inflammasome pathway due to genetic mutations (i.e., nucleotide-binding oligomerization domain, Leucine-rich Repeat and Pyrin domain 1/NLRP1, NLRP3, NLR family CARD domain-containing protein 4/NLRC4 or Pyrin), leads to common clinical phenotypes involving fever, skin rash, or dermatologic lesions and systemic symptoms (such as arthralgia or arthritis for example), associated with elevated inflammatory markers in the blood of affected patients [11]. For instance, NLRP1 mutations lead to NLRP1-associated Auto-Inflammation with Arthritis and Dyskeratosis (NAIAD) [12] and NLRC4 gain-of-function mutations triggers Syndrome of enterocolitis and Autoinflammation associated with mutation in NLRC4 (SCAN4) [13] or recurrent Macrophage Activation Syndrome (MAS) [14], (see Table 2).

Diseases induced by mutation in NLPR3 correspond to a group named Cryopyrin-Associated Periodic Syndrome (CAPS). CAPS represents a spectrum of related clinical phenotypes due to sporadic or autosomal dominant mutations in NLPR3 and previously described as three distinct diseases: NOMID (Neonatal-Onset Multisystem Inflammatory Disease, also known as Chronic Infantile Neurological Cutaneous Articular syndrome or CINCA), FCAS (Familial Cold Autoinflammatory Syndrome) and MWS (Muckle-Wells Syndrome) [15].

Diseases associated with Pyrin mutations encompass the Familial Mediterranean Fever (FMF), which is due to a gain-of-function mutation in the MEFV gene encoding Pyrin [16]. S242R and E244K mutations in pyrin sequence disrupt its interaction with the regulatory 14-3-3 protein and leads to the inflammasome activation and the constitutive IL1β and IL-18 secretion in a monogenic pathology called Pyrin-Associated Autoinflammation with Neutrophilic Dermatosis (PAAND) [17]. Additionally, the related PAPA syndrome (for Pyogenic sterile Arthritis, Pyoderma gangrenosum, and Acne) implicates mutations (A230T or E250Q) in the phosphatase PSTPIP1 that better interact with Pyrin promoting IL-1β and IL-18 secretion [18].

All these DAMP-sensor mutations and their associated diseases are summarized in Table 2. Interestingly, the high systemic inflammatory symptoms/systemic clinical manifestations observed with these pathologies seem to be shared with those observed in mice exhibiting an uncleavable caspase-8 and deficient for the necroptotic factors mixed-lineage kinase domain-like protein (MLKL) or receptor-interacting protein kinase 3 (RIPK3) [19]. Indeed, these mice die to a systemic and exacerbated inflammatory response that can be abrogated by the elimination of one CD95L allele [19], suggesting that a molecular link might exist between CD95 stimulation and the inflammasome activation in certain pathophysiological contexts.

## 3. CD95/Fas

CD95 is a member of the tumor necrosis family receptors (TNF-Rs) and it is the prototype receptor to study the apoptotic signaling pathway. TNF-Rs are devoid of any enzymatic activity and necessitate protein-protein interactions (PPIs) to recruit enzymes such as proteases, kinases, and ubiquitin ligases to implement dynamic and complex signaling pathways.

At the plasma membrane, CD95 auto-aggregates as homotrimer independently of its natural ligand, CD95L (also known as FasL or CD178) [20,21,22,23]. This trimeric structure is mandatory to implement cell death and rapidly forms larger signaling platforms in the presence of its natural ligand [20]. CD95 engagement induces the recruitment of the adaptor protein FADD, which in turn aggregates pro-caspase-8 in a complex-designated death-inducing signaling complex (DISC) [24]. Beyond DISC formation and induction of the apoptotic signal, FADD and caspase-8 contribute to different complexes involved in the induction of necroptosis or pyroptosis (discussed below).

The switch between apoptosis and necroptosis has been discussed previously [25]. Briefly, ubiquitination of RIPK1 is a pivotal post-translational modification for the induction of the TNF-R1-mediated NF-κB activation [26,27] and its deubiquitination leads to the induction of cell death. The deubiquitinated RIPK1 is released from the death receptor and recruits TRADD, Fas-associated death domain (FADD), pro-caspase-8, and the long isoform of FLICE-like inhibitory protein (FLIP_L_) to trigger apoptosis [28]. In this complex, the caspase-8-mediated RIPK1 cleavage extinguishes the kinase activity. Degradation of c-IAP1 and c-IAP2 prevents the K63 ubiquitination of RIPK1 [29] and promotes the formation of another cellular complex in which FADD, pro-caspase-8, and FLIP_L_ associate to trigger apoptosis.

When caspase-8 is inactivated in these two complexes, the necrosome is formed. RIPK1 associates with RIPK3 to form this complex, in which RIPK3 phosphorylates MLKL to promote its plasma membrane distribution and the induction of necrosis [30,31,32]. Although CD95 engagement displays a broad range of cellular outcomes [33], whether such complexes occur in a CD95-dependent manner remains to be elucidated.

Interestingly, the elimination of FADD or caspase-8 in mice does not lead to hyperplasia but instead is responsible for embryonic lethality, and these mice can survive post weaning when a double-ko mouse is realized with one of the necroptosis-mediating genes, RIPK3 [34,35] or MLKL [36] pointing out the tight control of the necroptotic machinery by the apoptotic one. Interestingly, caspase-8^−/−^/RIPK3^−/−^ or caspase-8^−/−^/MLKL^−/−^ double-ko mice develop hyperinflammation and lymphadenopathy, a phenotype resembling that observed in CD95-deficiency Lpr mice strongly suggesting that these factors control an additional mechanism in which CD95 could be involved too.

### 3.1. Classical Apoptotic Program

In the presence of CD95L, CD95 death domain (DD) recruits the adaptor protein FADD through homotypic interaction [37,38]. Then, the death effector domain (DED) of FADD binds the caspase-8 DED1 and the complete activation of the protease will require two molecular steps, (i) the oligomerization of caspase-8 via DED2/DED1 interaction [39,40], and (ii) its autocleavage to release a cytosolic active caspase. Mice lacking caspase-8 [34,35] or its activity (caspase-8 mutant C362S) [41,42] die during gestation because of the loss of the apoptotic control over the necroptotic process. Interestingly, caspase-8 mutants that fail to auto-aggregate or to undergo auto-cleavage do not trigger a strong CD95-mediated apoptotic signal but still exert a control over necroptosis through a residual caspase-8 activity [19]. Therefore, mice expressing a caspase-8 D387A (DA, uncleavable) or caspase-8 FGLG (mutations in the DED1 leading to no auto-aggregation) can block necroptosis during embryogenesis and are viable [19].

In agreement with the abrogation of the CD95-mediated apoptotic signal in these mice, caspase-8 DA or caspase-8 FGLG mice are resistant to fulminant hepatitis induced by injection of the agonistic anti-CD95 antibody Jo2 [19]. Nonetheless, elimination of RIPK3 and MLKL in these mice unravels an inflammatory burst when they are injected with the anti-CD95 agonistic mAb Jo2 [19]. Therefore, CD95 engagement triggers a pro-inflammatory program when caspase-8 cleavage and necroptosis are abrogated. Another TNF-R member designated TRAIL-R activates a complex called FADDosome, containing FADD and caspase-8 that induces the expression of pro-inflammatory genes independently of the caspase activity [43,44].

From a molecular standpoint, immune cells exposed to CD95L form a FADDosome in which FADD recruits caspase-8. A decreased caspase-8 activity, probably through a c-FLIP-dependent mechanism, impinges on the necroptotic signal but still allows the induction of a RIPK1-dependent inflammatory signal [19]. Interestingly, the kinase activity of RIPK1 was not required to secrete the inflammatory cytokines in this context. This original signaling pathway contributes to an inflammatory response via a process independent of caspases-1 and -11 activities [19].

Of note, caspase-8 cleavage site between its large and small catalytic subunits correspond to the amino acid residues L384/E385/V386/D387 and elimination of E385 (ΔE385) within this sequence generates an uncleavable caspase-8 [45]. Like caspase-8 DA/Mlkl^−/−^ mice [19], the hematopoiesis in Casp8ΔE385/Ripk3^−/−^ or Mlkl^−/−^ mice is skewed toward myeloid development [45]. The inflammatory response in Casp8ΔE385/Ripk3^−/−^ mice also relies on the scaffold property of RIPK1 [45] confirming that caspase-8 and RIPK1 form, in apoptotic altered and necrotic deficient immune cells, a scaffold involved in hematopoiesis and in the induction of an inflammatory response (Figure 1, green complex). In the presence of TNF, Casp8ΔE385 fails to trigger apoptosis [45], and enhances necroptosis by preventing the cleavage of RIPK1 questioning whether the interplay between apoptosis, necroptosis and inflammation is similarly regulated between TNF and CD95L receptors. Overall, these observations suggest the existence of a molecular link between CD95, the FADDosome (FADD/RIPK1/Casp8) and a pro-inflammatory signal.

### 3.2. Necroptosis

On a morphological basis, cell death occurs by apoptosis, necrosis, or autophagy [46]. Necroptosis is a well-regulated mechanism inhibited by Necrostatin-1 (Nec-1), which abrogates the kinase activity of RIPK1 [47]. The caspase-8 counteracts necroptosis by cleaving different necrotic factors such as RIPK1, RIPK3 and mixed-lineage kinase domain-like protein (MLKL) [48]. Upon different stimuli including TNF, CD95L, or IAP (inhibitor of apoptosis proteins) antagonists, the inhibition of caspase 8 (i.e., using chemical inhibitors or viral proteins) leads to the association of RIPK1 and RIPK3, their autophosphorylation, and transphosphorylate forming microfilament-like complexes [49] called necrosomes [50]. Although RIPK3 directly phosphorylates RIPK1, the opposite is not true [51] and the RIPK1 kinase activity seems instead responsible for stabilizing the necrosome [51]. The RIP homotypic interaction motifs (RHIMs) of RIPK1 and RIPK3 contribute to the necrosome formation [49]. RIPK3 is mandatory for necroptosis by phosphorylating MLKL and triggering its homotrimerization [31]. RIPK3 phosphorylates MLKL at the T357 and S358 [52] to induce its translocation to the plasma membrane and the disruption of cell membrane integrity [30,31]. At the plasma membrane, MLKL aggregation contributes to the activation of calcium channel TRPM7, which in turn mediates Na^+^, Ca^2+^, Mg^2+^, and Zn^2+^ influxes [32] (Figure 1). The MLKL translocation into the inner leaflet of the plasma membrane breaks the cell integrity.

Of note, DAMPs released by necroptotic cells stimulate immune cells through ligation of pattern recognition receptors (PRRs) to secrete pro-inflammatory cytokines such as interleukin-1β (IL-1β) and IL-18, which subsequently activate host immune defenses against various pathogens. The maturation of IL-1β and IL-18 is mediated by cytosolic NOD-like receptors (NLRs) and HIN domain-containing family member AIM2, which associate with other proteins to form a large multimeric complex designated the inflammasome [53,54]. The most extensively studied inflammasome consists of NLRP3, the apoptosis-associated speck-like protein containing a CARD (ASC) and pro-caspase-1 [54]. In the NLRP3 inflammasome, NLRP3 recruits ASC, which aggregates pro-caspase-1, triggering its auto-cleavage and the release in the cytosol of an activated protease. Activated caspase-1 cleaves and matures the pro-inflammatory cytokines IL-1β and IL-18 [55]. The inflammasome necessitates two signals, designated signals 1 and 2, to trigger the release of mature IL-1β and IL-18 [56]. Signal 1 leads to the NF-κB activation and the transcriptional upregulation of the inflammasome machinery and pro-IL-1β and pro-IL-18. Signal 2 is mandatory to promote the assembly of the inflammasome machinery in the cytosol and the subsequent caspase-1 activation releasing mature IL-1β and IL-18. Signal 2 is provided by a broad range of stimuli (i.e., DAMPs) including ATP and ions such as Cl^-^ and K^+^ effluxes [57] or Ca^2+^ influx [58]. Many links exist between necrosome and inflammasome since MLKL and RIPK3 can also contribute to the NLRP3 inflammasome activation by inducing signal 1, which is NFκB activation [59]. Although CD95 has been associated with necroptosis [60], a robust link between CD95 engagement and necrosome still remains to be established.

### 3.3. Non-Apoptotic Signals

The five members of the NF-κB family, consisting of RelA (p65), RelB, c-Rel, NFκB1 (p105), and NFκB2 (p100), share a conserved Rel homology domain (RHD) responsible for DNA binding. While p65, RelB, and c-Rel encompass a transactivation domain in their C-terminal regions, NFκB1 (p105) and NFκB2 (p100), which are degraded into p50 and p52, respectively, are devoid of transactivation domains [61]. Therefore, nuclear accumulation of p50/p50 homodimers might exert a transcriptional repression over the NF-κB response [61,62,63,64,65,66]. Stimulation of CD95 in cancer cells resistant to the apoptotic pathway triggers cell migration by inducing NF-κB signals [67,68,69]. CD95 can also induce a PI3K response that promotes cell migration [70,71,72,73,74,75,76,77] and endothelial transmigration of inflammatory neutrophils or Th17 T cells [78,79,80].

We recently established that the CD95 expression in triple-negative breast cancer (TNBC) cells is responsible for the partial degradation of p105 into p50 via the ubiquitin ligase KPC2/KPC1 [81]. The C-terminal region of CD95 directly binds KPC2, which in turn recruits KPC1 [81], and the loss of CD95 in TNBC cells impinges on this KPC2/KPC1-mediated p105 ubiquitination, leading to a decrease in p50. The p50 drop favors the formation of active p50/p65 heterodimers at the expense of p50/p50 homodimers and the induction of the pro-inflammatory NFκB response in CD95-deficient TNBC cells [81]. Interestingly, the induction of this pro-inflammatory signal seems to stimulate a natural killer (NK)-mediated anti-tumor response in TNBCs [82]. Overall, these findings suggest that CD95 expression in TNBCs behave as an immune checkpoint preventing the NK-mediated anti-tumor response.

Associated with c-FLIPL, the caspase-8 activity can induce the NF-κB signaling pathway in a complex consisting of caspase-8, c-FLIP_L_, RIPK1, and FADD to promote the differentiation of monocytes to macrophages in the presence of macrophage-colony-stimulating factor (M-CSF) [83]. The caspase-8-dependent cleavage of RIPK1 prevents the sustained NF-κB activation in this differentiation process. Although the death receptors do not seem to be involved in the monocyte differentiation [83], further investigation could address whether such a complex could account for the pro-inflammatory signal observed with non-cleavable caspase-8 [19]. In a similar manner, association of caspase-8 and c-FLIP_L_ favors CD8+ T-cell proliferation by recruiting RIPK1 and activating NF-κB [79]. From a molecular standpoint, c-FLIP_L_ is processed by caspase-8 into a p43 form that recruits RIPK1 to activate NF-κB [84]. CD95 could be involved in this process since pioneering studies highlighted that CD95 activation in T cells enhanced the CD3-mediated activation [85,86].

The activation of PI3K by CD95 engagement was described in the late 1990s [87], but its biological roles remained difficult to apprehend. Although initial reports described that CD95-mediated PI3K activation was crucial for ceramide synthesis and cell death, and this signal was induced by caspase, Ras [88], and p56Lck activities [89], contradicting findings highlighted that in fact, the CD95-mediated PI3K activation exerts an inhibitory function on caspase cleavage and apoptosis [90]. Although the CD95-mediated PI3K activation seems to rely on different members of the src kinase superfamily, how CD95 recruits these kinases remains difficult to apprehend. Epidermal growth factor receptor (EGFR) can phosphorylate caspase-8 at tyrosine 380 (Y380), in a src-dependent fashion, to inhibit the induction of the CD95-mediated apoptotic signal [91]. Although the protease activity is prevented, Y380 phosphorylation promotes cell migration by recruiting the p85α-regulatory subunit of phosphatidylinositol 3-kinase through its SH2 domain [92]. CD95 can associate with the receptor tyrosine kinases (RTK) EGF-R in cancer cells [74,93], the integrin LFA-1 in neutrophils [78] or PDGFR-β in colon cancer cells [75] to activate src kinases and promote cell migration. Therefore, CD95 can trigger different complex signaling pathways by recruiting transmembrane RTKs [33], and how this recruitment occurs remains to be elucidated.

## 4. Inflammation

Inflammation after infection or injury relies on the detection of pathogen-associated and damage-associated molecular patterns (PAMPs/DAMPs). Examples of sterile DAMPs includes cholesterol crystals (atherosclerosis), β-amyloid (Alzheimer’s), islet amyloid polypeptide, ceramide, saturated fatty acids (type II diabetes), asbestos, silica dioxide (pulmonary fibrotic disorders), and monosodium urate (gout) [8,94]. Pattern recognition receptors (PRRs) translate the cell stress into proinflammatory signals.

The cytosolic PAMPs/DAMPs activate inflammasome, which in turn induces caspase-1 activity leading to the maturation of interleukin-1β (IL-1β) and IL-18 [95,96]. Inflammasome can also induce pyroptosis by cleaving gasdermin D (GsdmD) [97,98]. GsdmD forms plasma membrane pores responsible for the induction of cell death and the release of mature IL-1β and IL-18 [99,100] (Figure 1). The signal 2 activating NLRP3 is diverse and includes mitochondrial reactive oxygen species (ROS), potassium efflux, and/or chloride influxes [101] (Figure 2), pointing out that NLRP3 is a sensor for a broad spectrum of cellular disturbances.

### 4.1. Inflammasome

Inflammasome activation requires NF-κB induction (priming signal or signal 1) for the upregulation, among others, of NLRP3 and pro–IL-1β; next, a danger signal (signal 2) activates inflammasome. FADD and caspase-8 form the FADDosome to activate NF-κB [102]. The caspase-8 activity does not contribute directly to this signal, but it serves as a scaffold [103] (Figure 1). On the other hand, the cytoplasmic dsRNA sensor requires caspase-8 activity to relieve the inhibition of RIPK3 on the NLRP3 inflammasome assembly [103]. Although this study highlights the pivotal role of caspase-8 and FADD in the activation of inflammasome, whether the death receptors contribute to this process remains to be elucidated. Interestingly, activation of the NLRP3-mediated inflammasome in macrophages induces FADD secretion [104]. It is tempting to envision that this secretion corresponds to a negative feedback loop to dampen the FADD-dependent inflammatory response by reducing the intracellular quantity of the adaptor protein.

GsdmD is cleaved at D275 in humans (D276 in mice) by caspase 1 in the canonical pathway and in the noncanonical pathway by caspases 4/5 in humans and caspase-11 in mice (orthologue). Interestingly, caspase-8 can also cleave GsdmD at the same site, providing an alternative method of activation [105,106]. Caspase-8 can also cleave IL-1β [107].

It is noteworthy that pioneer studies with injection of CD95L-expressing tumor cells into wild-type mice provokes a massive neutrophil infiltration [108] through an IL-1α/β-dependent but caspase-1-independent mechanism [109]. The caspase-1-independent release of IL-1β by macrophages exposed to CD95L has been confirmed [109,110]. More recently, it has been reported that CD95 engagement activates caspase-8 in macrophages and dendritic cells, leading to the maturation of IL-1β and IL-18 [111].

Antagonists of apoptosis proteins (IAPs) promote CD95-mediated cell death signals (i.e., apoptosis and necroptosis) [112]. IAP inhibitors also trigger IL-1β maturation via the NLRP3 inflammasome and caspase-1, as well as via a caspase-8–dependent pathway [107]. Elimination of IAPs engenders the IL-1β maturation through a RIPK3-dependent mechanism [107]. Therefore, the molecular targets of IAP antagonists including IAP1, 2, and XIAPs proteins control a RIPK3- and caspase-8-dependent inflammasome [107]. In agreement with the tight interplay between necroptosis, apoptosis, and inflammation, cIAP1, cIAP2, and XIAP also inhibit the RIPoptosome, a complex consisting of FADD/RIPK1/RIPK3/Caspase-8/FLIP_L_ and involved in apoptosis and necroptosis [113,114]. Whether CD95 can contribute to these signals remains to be defined. Rendering more complex the manner how CD95 engagement could secrete IL-1β, it has been reported that CD95 induces IL-1β maturation via a FADD- and caspase-8-dependent but NLRP3-, caspase-1-, caspase-11- and ASC-independent process in Bone-Marrow-Derived Macrophages (BMDMs) [111].

A catalytically inactive caspase-8 in which cysteine 362 has been replaced by an alanine (C362A) fails to trigger apoptosis and thereby to inhibit necroptosis, but can still induce ASC specks and the caspase-1-dependent cleavage of GsdmD [42]. Accordingly, while Casp8^−/−^Mlkl^−/−^ mice are viable, C362A^+/+^/Mlkl^−/−^ mice die during the perinatal period, confirming that the caspase-8 scaffold contributes to a toxic signal independently of its apoptotic role or its anti-necroptosis function. Caspase-1, caspase-11, and their adaptor ASC contribute to this signal [42]. From a molecular standpoint, the inactive caspase-8 adopts a different conformation as compared to its active counterpart, enabling its pro-domain to engage ASC and promote a caspase-1 and caspase-11-mediated inflammatory signal [42]. The lethal phenotype of C362A^+/+^/Mlkl^−/−^ mice is not prevented by elimination of Nlrp3 [42], raising the question of the Nod-like receptor (NLR) responsible for inflammasome activation in these animals.

As aforementioned, although the CD95-mediated apoptotic signal in immune cells expressing an oligomerization-deficient or a non-cleavable caspase-8 mutant is impaired, the protease can still counteract the necroptotic signal [19]. Nonetheless, elimination of RIPK3 or MLKL in these mice unleashes an inflammatory response when CD95 is stimulated, suggesting that the necrosome and the caspase-8 activity control a CD95-dependent inflammatory response. This signal occurs through the FADDosome (FADD/RIPK1/caspase-8) formation [19]. In epithelial cells (i.e., intestinal barrier), the elimination of FADD in Casp8DA/Mlkl^−/−^ animals causes an inflammatory burst resulting in mouse death within 2 weeks of life [19]. Therefore, FADD promotes inflammation in immune cells and inhibits it in epithelial cells [19]. The inflammasome inhibited by FADD in epithelial cells probably encompasses NLRP6 [19].

### 4.2. PAMP Sensors and CD95 Crosstalk

Extensive cross-talks exist between cell death pathways, and the simultaneous induction of pyroptotic, apoptotic and necroptotic molecules led to the concept of PANoptosis. AIM2 inflammasome senses double-stranded DNA (dsDNA) [115,116]. Unlike classical AIM2 inducer (i.e., poly(dA:dT)), HSV1 and F. novicida infections induce a large complex containing AIM2, pyrin, ZBP1, ASC, caspase-1 and -8, RIPK1, RIPK3, and FADD that induces PANoptosis. The authors called this complex the AIM2 PANoptosis [117]. AIM2 activation in macrophages induces the over-expression of CD95L in an IL-1β-dependent mechanism. This process contributes to the elimination of T-cells after severe tissue injury, and thereby increases susceptibility for life-threatening infections [118]. These findings indicate that although a link between CD95 stimulation and inflammasome remains to be identified, the activation of inflammasome can, on the other hand, modulate the immune landscape by triggering the CD95-mediated apoptotic signal.

cGAS and Stimulator of Interferon Genes (STING) have been identified as the sensors of cytosolic dsDNA [119]. Upon binding to dsDNA cGAS (a cyclic GMP/AMP synthase) converts GTP and ATP to the cyclic nucleotide called cGAMP [120,121]. STING binds cGAMP undergoing a conformational change, thereby facilitating the phosphorylation of the transcription factor IRF3, and finally leading to the expression of Type I IFN genes [120,121]. Both CD95 and INF signatures have been involved in the progression of lupus, but whether a link exists between these two pathways remains to be elucidated. Although the elimination of the INFγR in MRL/Lpr mice (CD95 deficient mice) protected them from glomerulonephritis, it remains inefficient to prevent lymphadenopathy and accumulation of double-negative T-cell (B220+CD3+CD4-CD8-) [122]. Interestingly skin lesions were higher in the INFγR^−/−^ MRL/Lpr mice as compared to their MRL/Lpr counterparts, suggesting a protective role of this signal in the skin inflammation [122]. cGAS or STING elimination in different models of lupus, in which CD95/Fas has been involved, also suggests that these factors could protect from the pathology progression. These factors alleviated the clinical symptoms by preventing the endosomal TLR signaling [123]. These data indicate that while TLR7 and TLR9 and its downstream transcription factor IFN regulatory factor 5 (IRF5) contribute to the SLE pathogenesis [124,125,126], this could occur in a cGAS/STING-independent pathway.

### 4.3. Ion Regulation of Inflammation

#### 4.3.1. Ion Channels and Inflammasomes

Intracellular ion homeostasis is sensed by inflammasome signaling. Non-isosmotic conditions implement NLRP3- and NLRC4-mediated caspase-1 activation in LPS-primed macrophages [127]. Potassium (K^+^) [128] and chloride (Cl^−^) [129] effluxes and calcium (Ca^2+^) [58] influx represent different but probably interconnected mechanisms contributing to the osmotic regulation and inflammasome activation. The K^+^ channel TWIK2 (two-pore domain weak inwardly rectifying K^+^ channel 2) [130] is responsible for K^+^ efflux (Figure 2), which promotes the association of NEK7 with NLRP3, activating the NLRP3-inflammasome [57]. Leucine-rich repeat-containing protein 8A (LRRC8A) is a channel allowing the passage across the plasma membrane of chloride (Cl^−^) ions [131]. The volume-regulated anion channels (VRAC), formed by Leucine-rich repeat-containing protein 8 (LRRC8) heteromers, also contribute to the NLRP3 inflammasome activation in hypotonic medium, while it does not participate in DAMP-induced inflammasome [132]. Cl^−^ efflux seems to be essential for the formation of ASC specks, while K^+^ efflux promotes the cleavage and maturation of IL-1β [57]. Interestingly, although the DAMP-mediated NLRP3 activation does not rely on VRAC, chloride efflux still modulates this signal suggesting that other chloride channels are involved in the inflammasome activation [132]. The chloride intracellular channel (CLIC) protein family consists of six evolutionary conserved proteins (CLIC1–CLIC6) implicated in various cellular processes [133]. CLIC1, CLIC4, and CLIC5 are expressed in macrophages. CLIC 1 and 4 promote the NLRP3 inflammasome and the subsequent pyroptotic response in macrophages [134,135] (Figure 2). Interestingly, K^+^ efflux alters the mitochondrion functions and promotes ROS production, which in turn favors the translocation of CLICs from ER to the plasma membrane for the induction of Cl^-^ efflux [135].

Phospholipase C hydrolyzes phosphatidyl inositol-4,5-bisphosphate (PIP2) into inositol-1,4,5-trisphosphate (IP3), a second messenger, which binds to inositol-1,4,5-trisphosphate receptors (IP3-R) at the endoplasmic reticulum (ER). IP3 binding to the IP3 receptors releases calcium from ER stores, leading to a transient peak of free calcium in the cytosol [2,33], involved in the assembly of inflammasome [58,136] (Figure 2). Calcium channels TRPM2 [137] and TRPV2 [129] contribute to the sustained Ca^2+^ response necessary for NLRP3 inflammasome activation. Interestingly, TRPM2 also contributes to the inflammatory response induced by TNFα in macrophages [138] and the activation of these channels in the CD95-mediated signaling pathway remains to be evaluated.

#### 4.3.2. Ion Channels and CD95

Of note, cell volume modulation also affects the CD95-mediated signaling pathway, suggesting a possible interplay between death receptors and VRACs. CD95 has been reported to activate different ion channels including K^+^, Cl^−^, and Ca^2+^. CD95 engagement invokes the Cl^−^ efflux [139] in a ceramide- and p56Lck-dependent mechanism [140,141]. Nonetheless, the chloride channel involved in the CD95-mediated signaling pathway and its role in the induced signal remain to be identified.

Interestingly, lymphocytes exposed to CD95L undergo an inhibition of K^+^ channel Kv1.3 through the production of ceramide and activation of Src-like tyrosine kinases [142,143] suggesting that Cl^−^ and K^+^ channels are regulated in an opposite fashion by CD95, and thereby, whether an effect exists on the inflammasome remains difficult to apprehend.

CD95 engagement stimulates the calcium response. The PLCγ1-mediated IP3 production and the subsequent activation of IP3-Rs leads to the release of ER-stored Ca^2+^ (Figure 2). This reduction in Ca^2+^ in ER lumen is sensed by ER transmembrane proteins STIM1 and STIM2 that traffic to the plasma membrane to activate Orai channels, allowing a sustained Ca^2+^ influx from the extracellular space [144,145,146,147], a molecular mechanism designated store-operated calcium entry (SOCE). CD95 can directly interact with PLCγ1 [79,80] to trigger IP3-R activation and SOCE-dependent Ca^2+^ response [148] suggesting that at least the signal 2 exists when this death receptor is stimulated. Because CD95 can also trigger NF-κB [67,69], we could postulate that CD95 might stimulate signal 1 and the inflammasome.

## 5. Conclusions and Open Questions

In invertebrates, TNF receptors share with inflammasome the biologic property to fight against pathogens. Indeed, ancestors of the TNF/TNFR superfamily present in phylum Mollusca (e.g., oysters, mussels, and clams) contribute to the elimination of Gram-negative bacteria [149]. In the lancelet fish, Amphioxus, lipopolysaccharide challenge promotes the increase in TNF and TNFR expression suggesting that this family might contribute to the elimination of pathogens [150]. Elegant in vivo data highlight that a molecular link exists between CD95 engagement and activation of the inflammasome, but this occurs in a cellular context in which the caspase activity is dampened and the necroptotic signal is inhibited [19]. Characterization of the molecular mechanism leading to inflammasome activation upon CD95 engagement, and the patho-physiological roles of this signal have still to be elucidated. We can envision that some apoptotic and necroptotic resistant tumor cells or certain infected cells, in which both cell death signals are under control, could undergo this signal.

## Figures and Tables

**Figure 1 cells-11-01438-f001:**
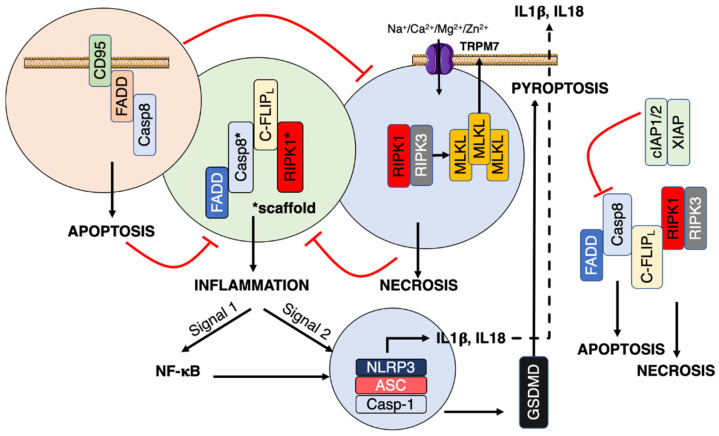
CD95 stimulation and potential links with inflammasome activation. At least three molecular complexes could occur upon CD95 engagement to trigger either apoptosis (**pink circle**), inflammation (**green circle**), or necrosis (**blue circle**). Apoptotic and necroptotic complexes control each other and the inflammatory complex (**red lines**). Necroptosis and pyroptosis signaling pathways lead to the break of the plasma membrane. IAPs also control an additional complex and these members are known to participate in the TNF-R-mediated signaling pathway. The NLRP3 inflammasome, which is activated by signal 1 + 2 is depicted. (GsdmD = Gasdermin D).

**Figure 2 cells-11-01438-f002:**
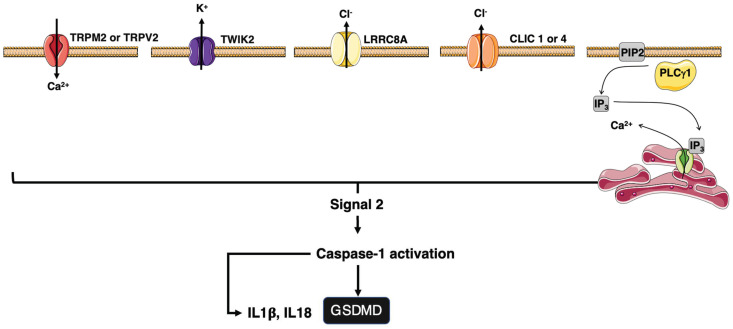
Representation of ion channels involved in the inflammasome activation. Different plasma membrane or endoplasmic reticulum ion channels have been involved in ion fluxes (**black arrows**) responsible for the signal 2 activating the inflammasome.

**Table 1 cells-11-01438-t001:** Auto-immune lymphoproliferative syndrome (ALPS) classification.

ALPS Type	Mutation Type
ALPS-FAS	CD95 germline homo- or heterozygous mutation
ALPS-sFAS	CD95 somatic mutation
ALPS-FASLG	CD95L germline mutation
ALPS-CASP10	Caspase 10 germline mutations
ALPS-U	ALPS phenotype with no known-ALPS mutation

**Table 2 cells-11-01438-t002:** DAMP-sensor-associated genetic disorders.

DAMP-Sensor Mutation	Mutation Type	Autoinflammatory Disorder
NLRP1	Loss of function	NAIAD
NLRP3	Gain of function	CAPS = NOMID, FCAS, MWS
NLRC4	Gain of function	SCAN4, MAS
Pyrin	Gain of function of MEFV or loss of function of MVK	FMF, HIDS, PAAND
PSTPIP1	Loss of function	PAPA

CAPS: Cryopyrin-Associated Periodic Syndrome; DAMP: Damage-Associated Molecular Patterns; FCAS: Familial Cold Autoinflammatory Syndrome; FMF: Familial Mediterranean Fever; HIDS: Hyper-IgD Syndrome; MAS: Macrophage Activation Syndrome; MVK: Mevalonate Kinase; MWS: Muckle-Wells Syndrome; NLRP: Nucleotide-binding oligomerization domain, Leucine-rich Repeat and Pyrin domain; NLRC4: NLR Family CARD Domain Containing 4; NOMID: Neonatal-Onset Multisystem Inflammatory Disease; PAAND: Pyrin-Associated Autoinflammation with Neutrophilic Dermatosis; PAPA, Pyogenic sterile Arthritis, Pyoderma gangrenosum, and Acne; PSTPIP1: Proline Serine Threonine Phosphatase-Interacting Protein 1; SCAN4: Syndrome of enterocolitis and Autoinflammation associated with mutation in NLRC4; TNFR1: TNF-Receptor-1; TRAPS: TNF-Receptor-Associated Periodic Syndrome.

## Data Availability

Not applicable.

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
