# Peer review of "Fas/CD95 Signaling Pathway in Damage-Associated Molecular Pattern (DAMP)-Sensing Receptors"

_cells, 2022, doi:10.3390/cells11091438_

Round 1
Reviewer 1 Report
This comprehensive review deals with the possible role of CD95/Fas signalling as a possible regulator of the DAMP-sensing and response.After listing the disorders linked to CD95 and DAMP genetic alterations, the authors describe in detail the signalling pathways controlled by CD95, and their alterations, and emphasize the crosstalks with PAMP/DAMP-sensing.
The text is clear, with adequate illustrations. The reference list is an adequate mix of classical, widely-accepted studies, and more recent and more discussed studies opening new investigation pathways.
As a conclusion, this is a timely and well-written review, that is certainly useful to investigators interested in this new aspects of CD95 signalling.
Author Response
We would like to thank the reviewer for his/her review.
Reviewer 2 Report
Reviewer´s report
Manuscript ID: cells-1659902
Title: Fas/CD95 signaling pathway in Damage Associated Molecular Pattern (DAMP)-sensing receptors
Version: 1
General comments:
In general, the review presents a relevant objective aimed to advance in the knowledge of CD95 signaling pathway and its role as a proinflammatory. Authors describe some of the links between CD95 signaling and DAMP sensors. For this purpose, they discuss CD95 stimulation in apoptosis, necroptosis, pyroptosis, or proinflammatory signaling pathways, and relates it’s with damage-associated molecular patterns (DAMPs).
Important Changes before publication:
- The first of these, and super important, is that the authors have not included the references section in their review. It is vitally important that this section is reflected, since without it is impossible to review the literature that these authors have chosen and there are some aspects that I have not been able to corroborate for this reason. They should resubmit their article in its complete form.
- In general, the review has a lot of information on molecular pathways and their targets. However, for a better understanding due to the large amount of information provided, the authors should detail the normal pathway with an outline and the alterations that occur in this route, in this way everything would be more organized. In most of the sections it is difficult to follow the thread for this reason.
- Some keywords do not correspond to those cited in the abstract. Please, correct it. When they are corrected, they should be ordered alphabetically.
- Introduction section is rather an index of the review or the objective of the review.
When the authors correct these changes, we reviewers will be able to focus on specific aspects of the review.
Author Response
Reviewer 2
General comments:
In general, the review presents a relevant objective aimed to advance in the knowledge of CD95 signaling pathway and its role as a proinflammatory. Authors describe some of the links between CD95 signaling and DAMP sensors. For this purpose, they discuss CD95 stimulation in apoptosis, necroptosis, pyroptosis, or proinflammatory signaling pathways, and relates it’s with damage-associated molecular patterns (DAMPs).
Important Changes before publication:
- The first of these, and super important, is that the authors have not included the references section in their review. It is vitally important that this section is reflected, since without it is impossible to review the literature that these authors have chosen and there are some aspects that I have not been able to corroborate for this reason. They should resubmit their article in its complete form.
Sorry about this missing! All references have been added.
- In general, the review has a lot of information on molecular pathways and their targets. However, for a better understanding due to the large amount of information provided, the authors should detail the normal pathway with an outline and the alterations that occur in this route, in this way everything would be more organized. In most of the sections it is difficult to follow the thread for this reason.
In the new version of the manuscript, we insisted on the “classical” apoptotic signal induced by CD95 in the main text. However, keep in mind that by now, how the switch occurs between apoptosis, necroptosis and now inflammation [1] in the CD95-mediated signaling pathway remains to be elucidated.
Moreover, we don’t really know if the apoptotic pathway is the “normal” pathway. Further investigations will be required to address if the different pathways are implemented simultaneously in cells exposed to CD95L or if these signaling pathways require different threshold of CD95 aggregation to be triggered as previously observed for the apoptotic and NF-kB signaling pathways [2,3].
- Some keywords do not correspond to those cited in the abstract. Please, correct it. When they are corrected, they should be ordered alphabetically.
We modified the keywords to reflect more specifically the abstract and the main text.
We also placed them according to an alphabetical order.
- Introduction section is rather an index of the review or the objective of the review.
You are right, introduction is more a short index of the review.
When the authors correct these changes, we reviewers will be able to focus on specific aspects of the review.
Sorry, for the fact that references were missing. Thank you for helping us to improve this review on CD95 and the potential links with the inflammatory pathways.
Reviewer 3 Report
Authors Gael Galli et al made a review of the literature concerning the link between the FAS signaling pathway and the Damage Associated Molecular Pattern (DAMP)-sensing receptors. They describe some of the existing links between CD95 signaling and DAMP sensors and propose that CD95 could act as an alarmin for pattern recognition receptors (PRRs).
Major comments:
- The citations are missing in the manuscript.
- Lines 99 to 103: the authors claimed that “the high systemic inflammatory symptoms observed in DAMP-sensor associated genetic disorders seem to be shared with those observed with CD95 mediated disease”; However, ALPS-FAS patients carrying mutation in the gene encoding for the death receptor do not present with any inflammatory syndrome. The mice models are confusing because of the genetic background of the MRL-Lpr that is responsible of the auto-inflammatory features.
- A table summarizing the DAMPs released by cell death and the main PRRs involved in immune system would be highly appreciated (ref to Amarente-Mendes et al, Frontiers in Immunology 2018).
- The introduction must be changed, as no clue in the text indicated that FAS could act as an alarmin for PRRs
Minor comments:
- The abbreviations (as PRRs-line 36) had to be explained when the authors used them at first, and only at first (FADD line 134).
- The authors cite some pathologies close to ALPS; they chose CASPASE-8, NRAS/KRAS and SH2D1a defects. They have to justify this choice, because other genetic deficiencies are associated with benign chronic lymphoproliferation (CTLA4; LRBA; STAT-3 GOF...).
- Lines 56 to 63: Mice with Fas mutation only develop clinical manifestations when they have an autoimmune genetic background (MRL). Black6 mice with the same Fas mutation develop lymphoproliferation and increased DN T cells, but do not develop autoimmunity.
- Line 158: authors refer to Figure 1 to illustrate the release of activated Caspase-8. The Figure 1 do not illustrate that.
- Line 226: Authors refer to the Signal 1: they have to add “(Figure 1)”, and better define what they called “signal 1” and signal 2 in the legend and in the text.
- Line 300: Authors explain that signal 2 could be diverse, but lead to NLRP3 activation. NLRP3 is not depicted neither in Figure 1 or Figure2.
Author Response
Reviewer 3
Authors Gael Galli et al made a review of the literature concerning the link between the FAS signaling pathway and the Damage Associated Molecular Pattern (DAMP)-sensing receptors. They describe some of the existing links between CD95 signaling and DAMP sensors and propose that CD95 could act as an alarmin for pattern recognition receptors (PRRs).
Major comments :
- The citations are missing in the manuscript.
As mentioned for reviewer 2, we are sorry about this missing.
This has been corrected.
- Lines 99 to 103: the authors claimed that “the high systemic inflammatory symptoms observed in DAMP-sensor associated genetic disorders seem to be shared with those observed with CD95 mediated disease”; However, ALPS-FAS patients carrying mutation in the gene encoding for the death receptor do not present with any inflammatory syndrome. The mice models are confusing because of the genetic background of the MRL-Lpr that is responsible of the auto-inflammatory features.
You are right. No inflammatory symptoms are observed in ALPS-FAS patients.
The sentence has been corrected to reflect the phenotype observed in mice unable i) to cleave caspase-8 and ii) to trigger necroptosis (Casp8DA/DAMlkl-/- and Casp8DA/DARipk3-/- mice) [1].
These mice die by developing a systemic inflammatory symptom, which is abrogated when one allele of CD95L is eliminated [1]. These findings strongly suggest that CD95 engagement in these mice is responsible for the induction of the inflammatory symptom, which is similar to that observed in animals with DAMP-sensor associated genetic disorders (gain of function mutants).
The sentence has been corrected.
- A table summarizing the DAMPs released by cell death and the main PRRs involved in immune system would be highly appreciated (ref to Amarente-Mendes et al, Frontiers in Immunology 2018).
Thank you for the reference.
The Frontiers in Immunology manuscript is very exhaustive in term of DAMPs and because the main topic of our submitted review is not on DAMPs, we decided to add this reference in the main text instead of a new table.
- The introduction must be changed, as no clue in the text indicated that FAS could act as an alarmin for PRRs
As requested by the reviewer, we have modified the sentences on CD95 and alarmin.
Minor comments:
- The abbreviations (as PRRs-line 36) had to be explained when the authors used them at first, and only at first (FADD line 134).
We have defined abbreviations at their first occurrence in the main text.
- The authors cite some pathologies close to ALPS; they chose CASPASE-8, NRAS/KRAS and SH2D1a defects. They have to justify this choice, because other genetic deficiencies are associated with benign chronic lymphoproliferation (CTLA4; LRBA; STAT-3 GOF...).
You are right.
Other genetic defects are associated with lymphoproliferation. We chose to cite these 3 pathologies because they have been reported in the classification of ALPS-related pathologies [4,5]. Of note, this classification also mention a 4th disease, designated DALD (Dianzani autoimmune lymphoproliferative disease), where the genetic deficiency is not yet identified.
We modified the text to be more accurate and reflect the current ALPS classification.
- Lines 56 to 63: Mice with Fas mutation only develop clinical manifestations when they have an autoimmune genetic background (MRL). Black6 mice with the same Fas mutation develop lymphoproliferation and increased DN T cells, but do not develop autoimmunity.
You are right, studies of Faslpr-congenic mice have established that the clinical characteristics and severity of disease depend on background genes.
Among the various strains of FasLpr mice, the MRLFaslpr develops the most severe autoimmune disease, with early onset autoantibody production, glomerulonephritis (GN), systemic vasculitis, arthritis, and 50% mortality at 5.5 months.
Insertion of the FasLpr mutation in C57BL/6 (B6) strain exhibits much milder symptoms, characterized by a lower lymphocyte accumulation, late onset autoantibody production, and none or minimal histopathologic manifestations [6]. In addition, there are several characteristics that distinguish MRLFaslpr mice from other lupus strains, including massive accumulation of DN B220+ T cells, high levels of serum immunoglobulins, a wide spectrum of autoantibodies that include anti-Sm and rheumatoid factor, and inflammatory arthritis.
These data confirmed i) that lupus is a multiparameter disorder and ii) that Fas mutation accelerates the pathology but alone, it does not trigger all symptoms observed in lupus patients [6].
In agreement with these statements, in addition to CD95 mutation, a genetic approach intercrossing MRL-Faslpr (severe disease) with C57BL/6-Faslpr (minimal disease) revealed four loci associated with lymphadenopathy and/ or splenomegaly and were designated lupus in (MRL-Faslpr x B6-Faslpr)F2 cross1-4 (Lmb1-4) [6].
- Line 158: authors refer to Figure 1 to illustrate the release of activated Caspase-8. The Figure 1 do not illustrate that.
Modified, thank you.
- Line 226: Authors refer to the Signal 1: they have to add “(Figure 1)”, and better define what they called “signal 1” and signal 2 in the legend and in the text.
We added a short text to explain the signal 1 and 2 required to activate the inflammasome in the main text and modified the figure 1.
- Line 300: Authors explain that signal 2 could be diverse, but lead to NLRP3 activation. NLRP3 is not depicted neither in Figure 1 or Figure2.
We added NLRP3 inflammasome (NLRP3/ASC/Casp-1) in Figure 1.
References
- Tummers, B.; Mari, L.; Guy, C.S.; Heckmann, B.L.; Rodriguez, D.A.; Ruhl, S.; Moretti, J.; Crawford, J.C.; Fitzgerald, P.; Kanneganti, T.D.; et al. Caspase-8-Dependent Inflammatory Responses Are Controlled by Its Adaptor, FADD, and Necroptosis. Immunity 2020, 52, 994-1006 e1008, doi:10.1016/j.immuni.2020.04.010.
- Lavrik, I.N.; Golks, A.; Riess, D.; Bentele, M.; Eils, R.; Krammer, P.H. Analysis of CD95 threshold signaling: triggering of CD95 (FAS/APO-1) at low concentrations primarily results in survival signaling. J Biol Chem 2007, 282, 13664-13671, doi:M700434200 [pii]
10.1074/jbc.M700434200.
- Legembre, P.; Barnhart, B.C.; Zheng, L.; Vijayan, S.; Straus, S.E.; Puck, J.; Dale, J.K.; Lenardo, M.; Peter, M.E. Induction of apoptosis and activation of NF-kappaB by CD95 require different signalling thresholds. EMBO Rep 2004, 5, 1084-1089, doi:10.1038/sj.embor.7400280.
- Lenardo, M.J.; Oliveira, J.B.; Zheng, L.; Rao, V.K. ALPS-ten lessons from an international workshop on a genetic disease of apoptosis. Immunity 2010, 32, 291-295, doi:10.1016/j.immuni.2010.03.013.
- Oliveira, J.B.; Bleesing, J.J.; Dianzani, U.; Fleisher, T.A.; Jaffe, E.S.; Lenardo, M.J.; Rieux-Laucat, F.; Siegel, R.M.; Su, H.C.; Teachey, D.T.; et al. Revised diagnostic criteria and classification for the autoimmune lymphoproliferative syndrome (ALPS): report from the 2009 NIH International Workshop. Blood 2010, 116, e35-40, doi:10.1182/blood-2010-04-280347.
- Vidal, S.; Kono, D.H.; Theofilopoulos, A.N. Loci predisposing to autoimmunity in MRL-Fas lpr and C57BL/6-Faslpr mice. J Clin Invest 1998, 101, 696-702, doi:10.1172/JCI1817.
Round 2
Reviewer 2 Report
The authors have improved their work and made the changes proposed by the reviewers
Reviewer 3 Report
Thank you for having take in account the suggestions/modifications I asked. Congratulation for your work.